# DMSO-Quenched H/D-Exchange 2D NMR Spectroscopy and Its Applications in Protein Science [note 1]

**DOI:** 10.3390/molecules27123748

**Published:** 2022-06-10

**Authors:** Kunihiro Kuwajima, Maho Yagi-Utsumi, Saeko Yanaka, Koichi Kato

**Affiliations:** 1Department of Physics, School of Science, University of Tokyo, 7-3-1 Hongo, Bunkyo-ku, Tokyo 113-0033, Japan; 2Exploratory Research Center on Life and Living Systems and Institute for Molecular Science, National Institutes of Natural Sciences, 5-1 Higashiyama, Myodaiji, Okazaki 444-8787, Aichi, Japan; mahoyagi@ims.ac.jp (M.Y.-U.); saeko-yanaka@ims.ac.jp (S.Y.); 3Department of Functional Molecular Science, School of Physical Sciences, SOKENDAI (the Graduate University for Advanced Studies), 5-1 Higashiyama, Myodaiji, Okazaki 444-8787, Aichi, Japan; 4Graduate School of Pharmaceutical Sciences, Nagoya City University, 3-1 Tanabe-dori, Mizuho-ku, Nagoya 467-8603, Aichi, Japan

**Keywords:** hydrogen/deuterium exchange, dimethylsulfoxide, nuclear magnetic resonance

## Abstract

Hydrogen/deuterium (H/D) exchange combined with two-dimensional (2D) NMR spectroscopy has been widely used for studying the structure, stability, and dynamics of proteins. When we apply the H/D-exchange method to investigate non-native states of proteins such as equilibrium and kinetic folding intermediates, H/D-exchange quenching techniques are indispensable, because the exchange reaction is usually too fast to follow by 2D NMR. In this article, we will describe the dimethylsulfoxide (DMSO)-quenched H/D-exchange method and its applications in protein science. In this method, the H/D-exchange buffer is replaced by an aprotic DMSO solution, which quenches the exchange reaction. We have improved the DMSO-quenched method by using spin desalting columns, which are used for medium exchange from the H/D-exchange buffer to the DMSO solution. This improvement has allowed us to monitor the H/D exchange of proteins at a high concentration of salts or denaturants. We describe methodological details of the improved DMSO-quenched method and present a case study using the improved method on the H/D-exchange behavior of unfolded human ubiquitin in 6 M guanidinium chloride.

## 1. Introduction

Hydrogen/deuterium (H/D) exchange is a powerful tool to investigate the structure, stability, dynamics, and interactions of proteins [1,2,3,4]. The advancements in two-dimensional (2D) nuclear magnetic resonance (NMR) techniques have made it possible to monitor the H/D-exchange kinetics of the individually identified peptide amide (NH) protons of proteins at amino-acid residue resolution [5,6]. Recent H/D-exchange studies also employ a mass spectrometric (MS) technique combined with rapid proteolysis and HPLC separation, which allows us to obtain the exchange kinetics at a nearly single-residue resolution [4,7,8]. In H/D-exchange experiments, the exchange reaction of each NH group of a protein to ND takes place in D_2_O. The exchange kinetics of the NH proton are determined by a structural opening reaction of the NH group and the intrinsic exchange rate constant, *k*_int_, of the NH proton, and are represented by:(1)$${\mathrm{NH}}_{\mathrm{cl}}\overset{k_{\mathrm{op}}}{\underset{k_{\mathrm{cl}}}{⇌}}{\mathrm{NH}}_{\mathrm{op}}\overset{k_{\mathrm{int}}}{⟶}\mathrm{ND},$$
where NH_cl_ and NH_op_ denote the closed and open states of the NH group, ND indicates the amide group after H/D exchange, and *k*_op_ and *k*_cl_ are the opening and closing rate constants [1,2,3,4]. Under the steady-state condition, the observed H/D-exchange rate constant, *k*_obs_, is thus given by:(2)$$k_{\mathrm{obs}}=\left(\frac{k_{\mathrm{op}}}{k_{\mathrm{op}}+k_{\mathrm{cl}}+k_{\mathrm{int}}}\right)\cdot k_{\mathrm{int}}.$$The *k*_int_ of each NH proton of a protein depends on solution conditions (pH, temperature, and salt concentration), neighboring residues (i.e., the amino-acid sequence), and isotope effects. Englander and his colleagues [9,10,11,12] accurately calibrated these effects on *k*_int_ and reported the calibration parameters, from which we can quantitatively estimate the *k*_int_ value of each NH proton of the protein.

The protection factor, *P*, which represents the degree of protection of the NH group against H/D exchange is given by the ratio of *k*_int_ to *k*_obs_ as:(3)$$P=\frac{k_{\mathrm{int}}}{k_{\mathrm{obs}}}.$$The *P* value of native globular proteins is as large as 10^6^–10^9^, which corresponds to the structural opening free energy of 8–13 kcal/mol. This free energy change is equivalent to the free energy change of the global unfolding for each protein, indicating that the H/D-exchange reactions of the most stable NH groups are brought about by global unfolding [13,14].

A native-state H/D-exchange method was developed by Bai et al. [15] in 1995, and the H/D-exchange behavior at low concentrations of denaturants was characterized by this method. Temperature or pressure perturbation was also used in the native-state H/D exchange [16,17,18]. The method is effective for studying different kinds of protein dynamic behavior, ranging from local fluctuations up to sub-global and whole-molecule global unfolding reactions [15,19,20,21]. The method has been applied to a large number of proteins (reviewed in [4,19,20,21]).

The H/D-exchange techniques have also been used effectively for studying non-native states, including equilibrium unfolding intermediates and transient intermediates in kinetic refolding reactions of proteins. The molten globule (MG) state, which has a substantial amount of secondary structure but lacks the specific tertiary side-chain packing characteristics of native proteins, is an equilibrium intermediate state under mildly denaturing conditions for numerous globular proteins, many with more than ~100 residues [22,23,24]. In the 1990s, the structural characterizations of the MG state by H/D-exchange 2D NMR were carried out for a number of globular proteins, including apomyoglobin [25], cytochrome *c* [26,27], α-lactalbumin [28,29,30,31,32], Ca^2+^-binding milk lysozyme [33,34], and other proteins [35,36,37,38,39,40,41]. The *P* values of slowly exchanging NH protons in the MG state range from 10^2^ to 10^3^, which is more than three orders of magnitude smaller than the values in the native (N) state. The H/D-exchange rate in the MG state and in other non-native unfolded states [42,43,44,45,46,47,48,49,50,51,52] is usually too fast to follow by 2D NMR spectroscopy. Therefore, the H/D exchange was quenched after the desired period of H/D exchange by rapid refolding, and the NMR spectra were measured in the N state.

The use of a hydrogen-exchange method to detect and characterize a transient folding intermediate of proteins was first reported by Schmid and Baldwin [53] in 1979. They used a tritium-exchange technique and carried out a kinetic competition between folding and hydrogen exchange to investigate the folding kinetics of ribonuclease A. Subsequently, Roder and Wüthrich [54] proposed an extension of this technique by combined use of rapid mixing and NMR analysis. In 1988, two seminal papers, one by Udgaonkar and Baldwin [55] for ribonuclease A and the other by Roder et al. [56] for oxidized cytochrome *c*, appeared and reported the structural characterization of kinetic folding intermediates by hydrogen-exchange labeling and 2D NMR spectroscopy. In both studies, the NH groups of proteins were first fully deuterated in the fully unfolded (U) state in D_2_O, and the nonprotected amide ND groups in folding intermediates were proton-labeled in H_2_O by a short alkaline pH pulse (pulse-labeling hydrogen exchange), followed by quenching the D/H exchange by rapid refolding to the N state and NMR measurements in the N state. Since then, the pulse-labeling or competition hydrogen-exchange studies combined with 2D NMR to detect and characterize transient folding intermediates have been reported for many globular proteins, including barnase by the Fersht group [57,58], the lysozyme–α-lactalbumin family proteins by the Chris Dobson group [59,60,61,62,63], apomyoglobin and apoleghemoglobin by the Wright group [64,65,66,67], and other proteins by other groups [41,68,69,70,71,72,73,74,75,76,77,78,79,80,81,82,83,84,85,86,87,88,89,90,91]. The early transient folding intermediates thus characterized are very similar in structure and stability to the equilibrium MG state, indicating that the MG state is the equilibrium counterpart of the kinetic folding intermediate formed early during refolding from the U state [23]. However, there are exceptions to this rule. The kinetic folding intermediate of the plant globin apoleghemoglobin under the refolding condition is significantly different in structure from its equilibrium MG state [66]. The early kinetic folding intermediate of ribonuclease H from *Escherichia coli* (*E. coli*) has a well-folded region with a closely packed tertiary structure, which is absent in its equilibrium MG state [92].

In this article, we describe the dimethylsulfoxide (DMSO)-quenched H/D-exchange method, in which the H/D-exchange reaction is quenched by the DMSO solution [93]. As shown above, the hydrogen-exchange reactions in the MG state, other non-native states, and transient kinetic folding intermediates of proteins need to be quenched before measurements of 2D NMR spectra. The rapid refolding was used for quenching the exchange reactions in the above studies, and hence, only NH protons stably protected in the N state are available for analysis. The DMSO-quenched H/D-exchange method is more versatile, because the exchange reactions of all NH groups, including nonprotected NH groups in the N state, are effectively quenched in an aprotic solvent DMSO and are available for the NMR analysis [93]. In the following, we will give a brief historical summary of the DMSO-quenched H/D-exchange method in proteins. In addition, the DMSO-quenched H/D-exchange method has recently been improved by the use of spin desalting columns [94]. This improvement has made it possible to apply the DMSO-quenched method to the exchange reactions of proteins in the presence of a high concentration of salt or denaturant. We thus describe the methodological details of the use of spin desalting columns in the DMSO-quenched H/D-exchange method and present a study on the H/D-exchange behavior of unfolded ubiquitin in 6 M guanidinium chloride (GdmCl), in which the spin desalting column was used in the DMSO-quenched H/D-exchange 2D NMR experiments.

## 2. DMSO-Quenched H/D-Exchange Method

The DMSO-quenched H/D-exchange method was developed by Zhang et al. [93] in 1995. They investigated various solution conditions to minimize the H/D-exchange rate of NH protons of proteins, and the presence of 95% (*v*/*v*) DMSO-*d_6_* in a DMSO-*d_6_*/D_2_O mixture at pH* 5–6 was the best condition, where the H/D-exchange rate was ~100 fold slower than the minimum exchange rate in D_2_O. Here, pH* is the uncorrected pH-meter reading, and the pH* was adjusted by dichloroacetic acid-*d_2_* (DCA-*d_2_*), whose p*K*_a_ is 5.72 in the DMSO/D_2_O mixture [93]. To use the DMSO-quenched H/D-exchange method to investigate the H/D-exchange reaction of a protein, aliquots of the reaction mixture at various exchange times are first quenched by rapid freezing in liquid nitrogen, and then lyophilized. The lyophilized powder is dissolved in the quenching DMSO solution, and the NMR spectra of the protein are measured. Because proteins are unfolded in the DMSO solution, the NMR peaks may be distributed in a very narrow spectral region. However, the use of isotope (^15^N and ^13^C)-enriched proteins and the triple-resonance multi-dimensional NMR techniques [95] have made it feasible to identify most of the cross peaks in the 2D ^1^H–^15^N heteronuclear single-quantum coherence (HSQC) spectrum [96] of a small protein and to follow the H/D-exchange kinetics of individually identified NH protons.

### 2.1. Applications to Folding Intermediates and Amyloid Fibrils

Nishimura et al. [65,66,97] applied the DMSO-quenched H/D-exchange method to elucidate the structure in the equilibrium and transient folding intermediates of apomyoglobin and apoleghemoglobin; they used 99.4% DMSO instead of the 95% DMSO solution as a quenching solvent, however. Using the DMSO-quenched H/D-exchange method, they acquired data for the NH protons of 94 residues for the 153 residues of apomyoglobin, as compared with the 52 residues probed by the conventional pulse-labeling hydrogen-exchange method, in which the exchange was quenched by rapid refolding [97]. The DMSO-quenched method could be applied only for pH-jump refolding experiments because it was difficult to dissolve the lyophilized protein in DMSO in the presence of residual denaturant (urea or GdmCl) after denaturant-jump experiments [65,66]. Sakamoto et al. [98] studied the H/D-exchange kinetics of disulfide-deficient lysozyme in glycerol solution by the DMSO-quenched H/D-exchange method. They removed glycerol from the reaction mixture by reversed-phase HPLC, and the fractionated portion was lyophilized before dissolving in the DMSO solution.

DMSO effectively dissolves amyloid fibrils in vivo and in vitro [99,100,101]. Therefore, since the early 2000s, the DMSO-quenched H/D-exchange method has been widely used in studies on the H/D-exchange kinetics of amyloid fibrils [102,103,104,105,106,107,108,109,110,111,112,113,114,115,116,117,118,119,120,121,122,123,124] and other protein aggregates, including protein supermolecular complexes [125,126]. These studies have demonstrated the presence of a hydrogen-bonded (H-bonded) core highly resistant to H/D exchange in the amyloid fibrils and the other protein complexes. In the amyloid experiments, insoluble amyloid fibrils were first suspended in D_2_O to carry out the H/D-exchange reaction for the desired exchange periods, followed by separation of fibrils by centrifugation and lyophilization. The lyophilized fibrils were dissolved and dissociated into monomers in the DMSO solution, and the unfolded monomeric form was subjected to 2D NMR analysis. To characterize transient kinetic intermediates during the formation of amyloid fibrils, Carulla et al. [127] and Konuma et al. [90] combined the pulse-labeling hydrogen-exchange strategy and the DMSO-quenched method. After a short labeling pH pulse, aliquots of the reaction mixture were frozen and lyophilized, followed by dissolution in the DMSO solution for the subsequent 2D NMR or MS analysis. These studies gave us information about the molecular mechanisms of amyloid formation. Although the DMSO solution based on the standard protocol is composed of 95% DMSO-*d_6_* at pH* 5–6 adjusted by DCA-*d_2_*, slightly modified compositions, e.g., dry DMSO-d_6_ [103,125] and DMSO-*d_6_*/trifluoroacetic acid-*d_1_* (0.01–1%) mixture [107,110,114,115,116,117,119,122,126,128,129] were also used as an H/D-exchange quenching solution. Several excellent review articles on the DMSO-quenched H/D-exchange method have been published and cover more details about the method [130,131,132,133,134].

### 2.2. Use of Spin Desalting Columns

We improved the DMSO-quenched H/D-exchange NMR method by the use of spin desalting columns for medium exchange from the H/D-exchange buffer to the DMSO solution [94]. As shown above, the conventional DMSO-quenched H/D-exchange experiments had used lyophilization for the medium exchange. Therefore, it was difficult to carry out the DMSO-quenched experiments at a high concentration of salt or denaturant (urea or GdmCl), because the presence of residual salt or denaturant after lyophilization interferes with the 2D NMR analysis of proteins dissolved in DMSO. This is a drawback of the conventional DMSO-quenched method, and it prevents us from using the method to characterize the H/D-exchange behaviors of proteins in the intermediate or unfolded state in denaturant and native proteins under physiological conditions at 0.15 M NaCl (or KCl).

To prepare the quenching DMSO solution, we adjusted the pH* of the 94.5% (*v*/*v*) DMSO-*d*_6_/5% (*v*/*v*) D_2_O/0.5% (*v*/*v*) DCA-*d*_2_ solution to between 5 and 6 by adding 10 M NaOD. The addition of NaOD was accompanied by crystalline sodium dichloroacetate-d_1_, which was dissolved by stirring with an increase in pH*. Before starting the H/D-exchange reaction of a sample protein, we first prepared a 10-fold concentrated stock solution of the protein in H_2_O. The H/D-exchange reaction was started by a 10-fold dilution of the stock solution with a D_2_O buffer. At appropriate time points in the H/D-exchange, we collected 1.0 mL aliquots of the reaction solution, which had been pre-dispensed into 2 mL polypropylene cryo-tubes in advance, and froze them by immersing the cryo-tubes in liquid nitrogen to stop the reaction. The frozen aliquots were kept at −85 °C to −80 °C until the medium exchange by a spin desalting column and the subsequent 2D NMR measurements. The frozen aliquots were thawed at an appropriate temperature, at or below room temperature, just before the medium exchange. The dead time of the H/D-exchange measurement depends on how the experiment is conducted. If the experiment is carried out with the cooperation of two experimenters, one quenching the reaction mixture quickly in liquid nitrogen and the other recording the reaction time of the H/D exchange, it is rather easy to realize the dead time of 30 s (see Figure 3).

Figure 1 shows a schematic of the medium exchange procedure from the D_2_O buffer of the H/D-exchange reaction mixture to the DMSO solution with the use of spin desalting columns [94]. We used 5 mL columns (Zeba^TM^ Spin Desalting Column 89891, Thermo Fisher Scientific K.K., Tokyo, Japan) and 15 mL polypropylene centrifuge tubes as collection tubes. The procedure consists of the following five steps:Remove the bottom plug of a spin desalting column, and place the column in a collection tube.Centrifuge for 2 min at 1000× *g* to pack the column bed and remove storage buffer.Apply 2.5 mL of the DMSO solution to the column, centrifuge for 2–3 min at 1000× *g*, and discard the flow through. Repeat this process two to three times, but centrifuge for 3 min in the second and third runs, because the DMSO solution is more viscous than water.Apply 1.0 mL of the H/D-exchange sample solution to the column, and centrifuge for 2 min at 20 °C. The application volume should not exceed 1.0 mL, because the application of more than 1.0 mL results in leakage of the H/D-exchange sample solution into the collection tube.Recover the desalted protein sample in the DMSO solution; the protein sample can now be applied to 2D NMR analysis.

**Figure 1 molecules-27-03748-f001:**
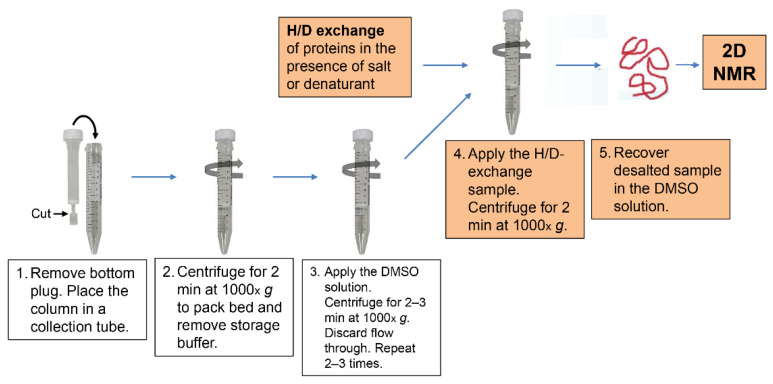
A schematic procedure of the medium exchange by a spin desalting column.

We used 2D ^1^H–^15^N HSQC spectra for the NMR analysis, and hence the sample protein was ^15^N-labeled, and the assignment of the HSQC cross peaks was carried out by three-dimensional (3D) HN(CA)NNH, HNCA, HN(CO)CA, HNCO, CBCA(CO)NH, and HNCAHA experiments using ^13^C/^15^N-double-labeled proteins. We applied the improved DMSO-quenched 2D NMR method to investigate the H/D-exchange behavior of the *E. coli* co-chaperonin GroES at pH* 6.5 (or 7.5) and 25 °C [135] and unfolded ubiquitin at pH* 3.2 and 15.0 °C in 6 M GdmCl [136], In the following, we will describe the study on the H/D-exchange behavior of unfolded ubiquitin as a case study.

## 3. A Case Study: Unfolded Ubiquitin in 6 M GdmCl

The characterization of residual structures persistent in unfolded proteins in concentrated denaturant (6 M GdmCl or 8 M urea) is an important issue in studies of protein folding. The problem of protein folding has been described with reference to the Levinthal paradox, in which the initial unfolded state is assumed to be a random coil, and hence, there may exist an astronomically large number of conformations, inaccessible in a reasonable time by a random search, at the beginning of the folding reactions [137,138,139]. Solving the Levinthal paradox is a fundamental problem in folding studies [140,141,142,143,144,145,146]. The presence of the residual structure, if any, in the unfolded state thus invalidates the Levinthal paradox, because such residual structure may form a folding initiation site and guide the subsequent folding reactions. We therefore studied the H/D-exchange behavior of unfolded human ubiquitin in 6 M GdmCl by the DMSO-quenched H/D-exchange 2D NMR method with the use of spin desalting columns [136]. Although the persistence of residual structures in unfolded proteins in concentrated denaturant has been reported for several different proteins [50,147,148,149,150,151,152,153], the present method enabled us to estimate the *p* values of individually identified NH protons, including nonprotected NH protons in the N state [136].

Ubiquitin is a 76-residue α/β protein, composed of a mixed parallel–anti-parallel β-sheet packing against a middle α-helix to form the hydrophobic core (Figure 2) [154]. Ubiquitin is a typical model protein for protein folding studies, and thus its folding reactions have been studied by a variety of biophysical techniques, including stopped-flow [155,156,157,158] and continuous-flow [159] kinetic refolding experiments, pulse-labeling hydrogen-exchange experiments combined with 2D NMR spectroscopy [72,160] and electrospray ionization mass spectrometry [161], mutational ϕ-value analysis [162,163], and other techniques [164,165,166]. These results may be compared with the present results of the H/D-exchange behavior of unfolded ubiquitin in 6 M GdmCl.

To analyze the H/D-exchange kinetics of individually identified NH protons of ubiquitin, we first made their spectral assignments in the DMSO solution [136]. Using a combination of 3D spectral measurements, we successfully assigned all the peaks observed in the HSQC spectrum. We then investigated the H/D-exchange kinetics of all the individual NH groups. Excluding NH groups whose residues could not be used as probes due to severe broadening or overlapping, we successfully followed the H/D-exchange kinetics of 60 NH protons [136]. These 60 NH protons include not only the protons stably protected in the native structure but also nonprotected NH protons. The observed kinetic exchange curve, given by the volume, *Y*(*t*), of cross peaks in 2D NMR spectra as a function of the H/D-exchange time, *t*, was a single exponential fitted to the equation:(4)$$Y(t)=\Delta Y\cdot e^{-k_{\mathrm{obs}}t}+Y(\infty ),$$
where Δ*Y* and *Y*(*∞*) are the kinetic amplitude and the final value of the peak volume, respectively. Figure 3 shows the kinetic progress curves of the Val5, Asn25, Gln40, and Glu51 NH protons measured by the DMSO-quenched method. The protection factor *P* was calculated by Equation (3) for the 60 NH protons, resulting in the protection profile shown in Figure 4, in which the *P* value is plotted as a function of the residue number.

From Figure 4, a majority of the NH protons have a *P* value below 2 and larger than or equal to 0.8, indicating that these peptide NH protons are almost fully exposed to solvent water in 6 M GdmCl. This is consistent with the previous reports that ubiquitin in concentrated denaturant (6 M GdmCl or 8 M urea) at acidic pH is almost fully unfolded [155,167,168,169,170]. However, the 10 NH protons of Asn25, Ala28, Lys33, Gln40, Arg42, Gln49, Glu51, Asp52, Glu64, and Ser65 were significantly protected with a *p* value larger than 3, and the additional 14 NH protons of Lys6, Thr7, Lys11, Val17, Glu18, Thr22, Lys27, Lys29, Asp32, Glu34, Leu43, Tyr59, Lys63, and Arg72 showed *p* values between 2 and 3. These results thus clearly indicate the presence of residual structures in unfolded ubiquitin. Because the protein was unfolded in 6 M GdmCl, it is most likely that these residues were protected by the formation of an H-bond with a certain acceptor group.

When the H/D-exchange protection is brought about by the formation of the H-bond with a specific acceptor group, we can estimate the fraction of H-bonding (*f*_Hbond_) for the protected NH groups. Because only the non-H-bonded form of the NH proton is available for H/D exchange, (1 − *f*_Hbond_) is equal to *k*_obs_/*k*_int_ (= 1/*P*). Therefore, it follows that:(5)$$f_{\mathrm{Hbond}}=1-\frac{1}{P}$$When NH protons have *P* values larger than 3 and 2, the *f*_Hbond_ values are larger than 0.67 and 0.50, respectively, from Equation (5). The free energy of the H-bond breakage (0.0–0.41 kcal/mol) estimated from the *f*_Hbond_ values is thus negligibly small as compared with the unfolding free energy (7.5 kcal/mol) of ubiquitin [155]. Nevertheless, the *f*_Hbond_ values larger than 0.5 should be significant when we consider the kinetic folding mechanisms of the protein, and some of these weakly protected residues may play an important role in the formation of folding initiation sites at an initial stage of kinetic refolding of the protein from the GdmCl-induced unfolded state.

To understand the relationships between the residual structure in unfolded ubiquitin and the H-bonds formed in native ubiquitin, the H-bonding network in the native structure is shown in Figure 5. From Figure 4 and Figure 5, the NH protons of Asn25, Lys27, Ala28, Lys29, Asp32, Lys33, and Glu34, which are all significantly protected with a *P* value larger than 2, are involved in the middle α-helix (Ile23–E34) in native ubiquitin, and each NH proton of these residues forms an α-helical H-bond with the peptide CO group of the four amino-acid residues earlier, except for Asn25 (Figure 5B). The Asn25 NH proton forms a more local H-bond with the side-chain O^γ^ atom of Thr22, which acts as an N-cap residue of the helix in native ubiquitin. These results thus clearly demonstrate the presence of a residual structure in this α-helix of ubiquitin in 6 M GdmCl at pH* 3.3 and 15 °C, and Thr22 may also function as a helix stop signal by forming the N-cap conformation as in the native ubiquitin structure [171]. The NH groups of Thr7 and Val17, which form H-bonds with the CO groups of Lys11 and Met1, respectively, in the N-terminal β-hairpin, also have *P* values larger than 2 in the U state (Figure 5C), suggesting that the N-terminal β-hairpin may also be partially preserved in unfolded ubiquitin. The other NH groups having a *P* value larger than 2 in the N-terminal β-hairpin include those of Lys6 and Lys11. Although the NH proton of Lys11 does not form a backbone H-bond in native ubiquitin, it forms a local backbone to the side-chain H-bond with the O^γ^ of Thr7, and hence such a local backbone to the side-chain H-bond may be at least partially preserved in unfolded ubiquitin and may stabilize the residual structure of the N-terminal β-hairpin (Figure 5C). Previous NMR studies on H^N^–N^N^ residual dipolar couplings, chemical shifts, ^3^*J*_HNHA_ couplings, relaxation rates, and ^h3^*J*_NC′_ couplings have also shown that the native-like first β-hairpin conformation was populated to at most 25% in unfolded ubiquitin in 8 M urea [172,173,174,175]. From these results, we conclude that there are native-like residual structures in the middle helix and the N-terminal β-hairpin in unfolded ubiquitin in 6 M GdmCl and that these residual structures may play an important role at an initial stage of kinetic refolding from the unfolded state.

In support of this conclusion, a pulsed H/D-exchange study with rapid mixing methods and 2D NMR analysis by Briggs and Roder [72] has shown that the NH protons in the α-helix and the β-hairpin of the fast-folding species (ca. 80%) of unfolded ubiquitin become protected in an initial 8 ms folding phase from the GdmCl-induced unfolded state of ubiquitin. Went and Jackson [163] performed a comprehensive ϕ-value analysis on the structure of the transition state ensemble of the ubiquitin folding and showed that medium and high ϕ values were found only in the N-terminal β-hairpin and the middle helix, which was also consistent with the above conclusion. The α-helical residual structure as detected by the unfolded-state H/D-exchange NMR spectroscopy in a concentrated denaturant solution was also observed in cytochrome *c* in 6 M urea [91], suggesting that the present observation of the residual structure may not be a rare example.

Three locally stabilized H-bonds, which form two one-turn 3_10_ helices (Gln40–Pro37 and Tyr59–Leu56) and a type II β-turn (Ser65–Gln62) in native ubiquitin, are also partially preserved in 6 M GdmCl. The NH proton of Gln40, which has a *P* value of 5.5 (*f*_Hbond_ = 0.81), forms H-bonds with the CO groups of Pro37 and Pro38 (Figure 5D). The NH proton of Tyr59, having a *P* value of 2.5 (*f*_Hbond_ = 0.60), forms an H-bond with the CO group of Leu56. The NH proton of Ser65, having a *P* value of 3.5 (*f*_Hbond_ = 0.71), forms an H-bond with the CO group of Gln62 (Figure 5E). Therefore, these locally stabilized H-bonds may not be fully disrupted even in 6 M GdmCl. However, it is not yet clear whether these local H-bonding interactions are important for the kinetic folding mechanisms of ubiquitin, because there is a dearth of experimental data concerning the effects of these local H-bonds on the kinetics of the ubiquitin folding. 

The NH protons of the three residues of Arg42, Glu64, and Arg72 form native H-bonds with the backbone CO groups of Val70, Gln2, and Gln40, respectively [154], and have *P* values of 2.7 to 6.5 in unfolded ubiquitin (Figure 4). However, these are nonlocal H-bonds formed between residues at least 28 residues apart from each other, and such nonlocal H-bonds in native ubiquitin may not be stably formed in the U state in 6 M GdmCl. These NH protons may thus be protected by non-native H-bonding interactions. In fact, certain other NH protons, including those of Leu43, Gln49, Asp52, and Lys63, which have *P* values of 2.1 to 7.5 (Figure 4), are protected by non-native H-bonding interactions in 6 M GdmCl, because these NH protons do in fact lack backbone H-bonds in the native structure. The majority of residues with a *P* value larger than 3 are either charged or contain a side-chain amide or guanidinium group (Arg42, Gln49, Glu51, and Glu64). It is thus also possible that the protection might be afforded by the H-bonding between the backbone NH and its own side-chain atoms.

## 4. Conclusions

We described the DMSO-quenched H/D-exchange 2D NMR spectroscopy and its applications in protein science. The DMSO-quenched method is superior to the conventionally used refolding-quenched method, because the former allows us to monitor the H/D-exchange kinetics of not only the protected NH protons but also the nonprotected NH protons in the N state.We described the improvement of the DMSO-quenched H/D-exchange method by the use of spin desalting columns and gave the methodological details of the use of spin desalting columns.We presented a case study in which we characterized the H/D-exchange kinetics of 60 peptide NH protons in unfolded ubiquitin in 6 M GdmCl by the improved DMSO-quenched method. The residual structures were preserved in the middle α-helix and the N-terminal β-hairpin in unfolded ubiquitin with a protection factor *P* larger than 2 (i.e., a fraction of H-bonding *f*_Hbond_ of larger than 0.5), and these residual structures may play an important role in the folding process as nucleation sites and guide the subsequent folding reactions to the N state.

## Figures and Tables

**Figure 2 molecules-27-03748-f002:**
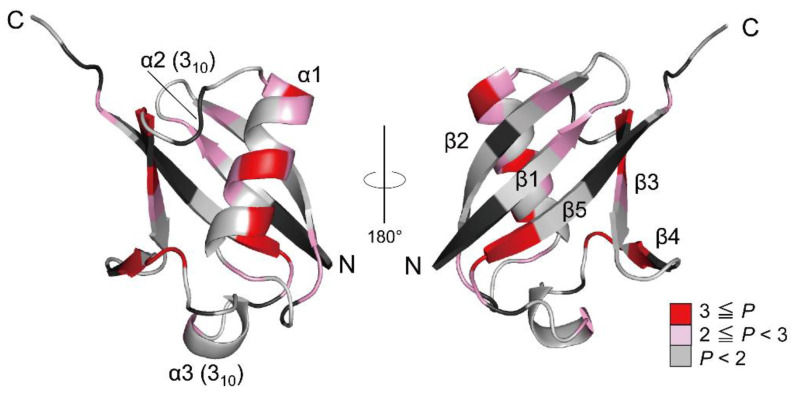
The 3D structure of native ubiquitin (PDB code: 1UBQ). The residues are colored according to the *P* values of the NH protons, and the red gradient indicates the scale of the *P* value. The proline residues and the residues that could not be used as probes due to severe broadening or overlapping are shown in black. Adapted with permission from Ref. [136]. Copyright 2020 Biophysical Society.

**Figure 3 molecules-27-03748-f003:**
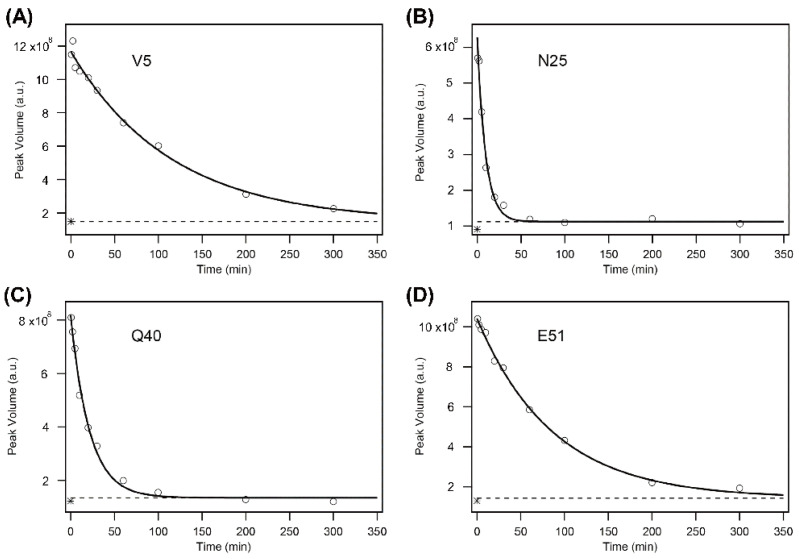
The H/D-exchange curves for Val5 (V5) (**A**), Asn25 (N25) (**B**), Gln40 (Q40) (**C**), and Glu51 (E51) (**D**) of human ubiquitin in 6 M GdmCl at pH* 3.3 and 15 °C. The solid lines are the theoretical curves best fitted to a single-exponential function (Equation (4)). A broken line in each panel indicates the theoretically estimated peak volume after the complete exchange (i.e., *Y*(∞) in Equation (4)), and an asterisk “*” in each panel, located between (1–2) × 10^8^ of the peak volume, indicates the experimentally observed value after heating the sample at 50 °C for 30 min. Because the reaction mixtures contained 10% H_2_O, the final peak volumes did not reach zero. The *k*_obs_ values for the four residues are: (**A**) (8.7 ± 0.8) × 10^−3^ min^−1^; (**B**) (10.5 ± 1.2) × 10^−2^ min^−1^; (**C**) (4.6 ± 0.4) × 10^−2^ min^−1^; (**D**) (1.2 ± 0.1) × 10^−2^ min^−1^. The first and last time points are at 0.62 and 300.09 min, respectively, for all panels. Adapted with permission from Ref. [136]. Copyright 2020 Biophysical Society.

**Figure 4 molecules-27-03748-f004:**
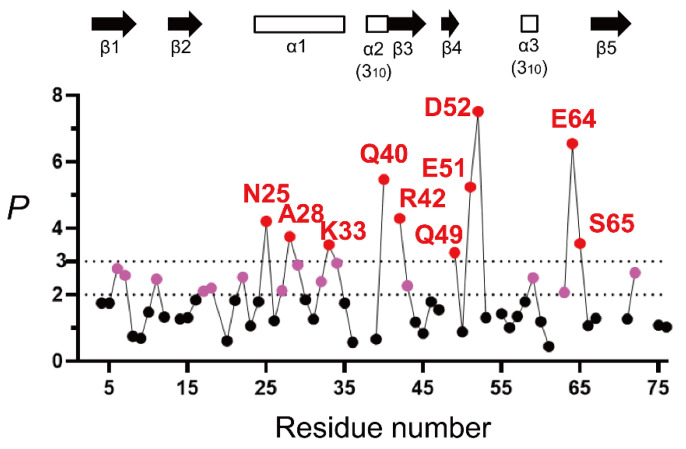
The H/D-exchange protection profile of unfolded ubiquitin in 6 M GdmCl, represented by *P* as a function of the residue number (pH* 3.3 and 15 °C). The dashed lines indicate the *P* values of 2 and 3. The amino-acid residues with *P* values larger than 2 and 3 are indicated in pink and red, respectively, and the other residues in black. The locations of the secondary structures in native ubiquitin (PDB code: 1UBQ) are shown by arrows (β-strands) and open rectangles (helices). The *k*_obs_ values for the majority of NH protons were obtained by three independent H/D-exchange experiments, and the percent standard error estimate of *k*_obs_ was ~8%, indicating that the *P* value of 2.0 can be written as *P* = 2.0 ± 0.16 (see [136] for the complete list of the standard error estimates of *k*_obs_ values). Adapted with permission from Ref. [136]. Copyright 2020 Biophysical Society.

**Figure 5 molecules-27-03748-f005:**
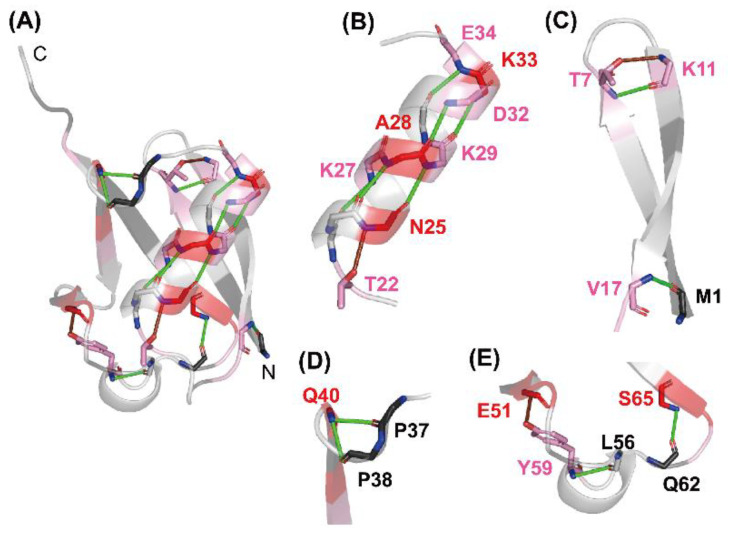
The H-bonding network observed in native ubiquitin (PDB code: 1UBQ). (**A**) A whole view, and (**B**–**E**) closer views of (**B**) the middle α-helix, (**C**) the N-terminal β-hairpin, (**D**) the one-turn 3_10_ helix (Pro38–Gln40), and (**E**) the Type II β-turn (Gln62–Ser65) and the one-turn 3_10_ helix (Ser57–Tyr59) are shown. The H-bonds of the NH protons of Thr7, Val17, Lys27, Ala28, Lys29, Asp32, Lys33, Glu34, Gln40, Tyr59, and Ser65 with the CO groups of their counterparts are shown as green lines. The local H-bonds formed by the NH protons of Lys11, Asn25, and Glu51 with the side-chain atoms of Thr7, Thr22, and Tyr59, respectively, are shown as brown lines. The red gradient indicates the same *P* value scale as shown in Figure 2. Adapted with permission from Ref. [136]. Copyright 2020 Biophysical Society.

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
