# Peer review of "DMSO-Quenched H/D-Exchange 2D NMR Spectroscopy and Its Applications in Protein Science†"

_molecules, 2022, doi:10.3390/molecules27123748_

Round 1

Reviewer 1 Report

It was a pleasure to read this review of solvent-quenched H/D-exchange methods. The use of non-aqueous solvents such as DMSO to effectively quench the H/D exchange reaction, combined with 2D NMR or mass spectrometry, has become an important tool for studies of protein folding, dynamics and interactions. While the set of observable amide groups in conventional quenched H/D exchange experiments is limited to those that are limited to those that are protected in the folded or ligand-bound state of the protein, this strategy makes it possible to measure exchange rates for a more complete set of amide probes, including those that are unprotected under quench conditions. Kuwajima and colleagues provide an excellent introduction to the principles and applications of hydrogen exchange, focusing on quenched H/D exchange protocols, including an effective spin-column desalting technique they developed recently. Two case studies nicely document the power of the method for characterizing oligomeric proteins and denatured protein states, respectively. The review is well organized and clear, and generally well written (except for a few minor grammatical errors). I recommend publication of the manuscript upon minor revision to address the following:

  1. While I realize that the uncertainty in the prediction of intrinsic exchange rates, based on the model-peptide data by Bai et al., makes a rigorous error analysis difficult, a qualitative discussion of errors in P-values would be helpful. In particular, some justification is needed for the assumption that P-values >2 are indicative of residual (H-bonded) structure. This is especially important for assessing the significance of protection factors that are only marginally larger than 2, such as those in the N-terminal segment.
  2. In your discussion of amide protection patterns in denatured ubiquitin (p. 11, paragraph starting on line 421), you might mention the possibility that a given NH group can dynamically engage multiple H-bond acceptors rather than forming one specific hydrogen bond, as stated on line 431.
  3. You may want to comment on the curious fact that the majority of residues with P>3 in Fig. 7 (8 out of 10) are either charged or contain side-chain amides. Is it possible that these residues can form local side-chain/backbone hydrogen bonds, which might contribute to solvent protection of their amide (either directly or by locally constraining backbone mobility? Similar trends were reported for cytochrome c in 6 M urea, using a rapid quenched-flow mixing technique (Fazelinia et al., J Amer Chem Soc 136, 733-740, 2014; PMCID: PMC3956590).

Author Response

Reviewer 1

  1. While I realize that the uncertainty in the prediction of intrinsic exchange rates, based on the model-peptide data by Bai et al., makes a rigorous error analysis difficult, a qualitative discussion of errors in P-values would be helpful. In particular, some justification is needed for the assumption that P-values >2 are indicative of residual (H-bonded) structure. This is especially important for assessing the significance of protection factors that are only marginally larger than 2, such as those in the N-terminal segment.

We thank the reviewer for the valuable comment. In the legend of Fig. 7 of the revised manuscript (lines 489–492), we have described an expected error estimate of the P value. The complete list of the standard error estimates of kobs values is given in the original article (Yagi-Utsumi et al. (2020) Biophys. J. 119, 2029–2038).

  1. In your discussion of amide protection patterns in denatured ubiquitin (p. 11, paragraph starting on line 421), you might mention the possibility that a given NH group can dynamically engage multiple H-bond acceptors rather than forming one specific hydrogen bond, as stated on line 431.

To avoid this uncertainty, we have changed the sentence (lines 470–471) as follows:

When the H/D-exchange protection is brought about by formation of the H-bond, we can estimate the fraction of H-bonding--->

When the H/D-exchange protection is brought about by formation of the H-bond with a specific acceptor group, we can estimate the fraction of H-bonding

  1. You may want to comment on the curious fact that the majority of residues with P>3 in Fig. 7 (8 out of 10) are either charged or contain side-chain amides. Is it possible that these residues can form local side-chain/backbone hydrogen bonds, which might contribute to solvent protection of their amide (either directly or by locally constraining backbone mobility? Similar trends were reported for cytochrome c in 6 M urea, using a rapid quenched-flow mixing technique (Fazelinia et al., J Amer Chem Soc 136, 733-740, 2014; PMCID: PMC3956590).

We thank the reviewer for this important suggestion. The possibility of local side-chain/backbone H-bonds is described in the revised manuscript (line 564–568). The paper by Fazelinia et al. (2014) was cited (ref. 91) and discussed as a similar observation of the residual α-helical structure (lines 541–544).

Reviewer 2 Report

The authors reviewed the applications of DMSO-quenched H/D exchange 2D NMR Spectroscopy to protein structure/folding analysis. One of the main application of this technique is to investigate non-native states of proteins as folding intermediates. This technique derives from the pulse-labeling technique proposed by Roder et al.  years ago, and brings actually significative improvements, especially when fast-exchanging protons (not involved in H-bonds in secondary structures) are concerned. Whether the use of this technique remains “anecdotic” in the literature (the two “case studies” described in the review come from their own group), the authors made a valuable effort to promote it. Based on a very large number of relevant references, the review is clearly written and easy to read, and deserve to be published after considering the following points.

- Whether pulse-labeling NMR techniques are efficient to highlight protein folding intermediates, there are others techniques that should be mentioned. Even if the authors want to restrict their review to the NMR methods (which is reasonable!), the considerable contribution of High-Pressure NMR to the field in the last decade should be cited. The authors can choose some seminal works like: Kitahara and Akasaka PNAS 100, 3167-72, 2003, or Roche et al., PNAS 109, 6945-50, 2012, or Fossat et al., BJ 111, 2368-76, 2016. Instead, they can also cite recent reviews like Roche et al., Prog. NMR Spectr. 102-103, 15-31, 2017 or Dubois et al., Molecules 25, 5551, 2020. Also, they must mention the work of Fuentes and Wand Biochemistry 37, 9877-9883, 1998, who combined H/D exchange and HP-NMR to investigate protein folding.

- This technique is supposed to be very efficient for fast-exchanging protons, but I cannot find any information on the time-scale covered, especially for the shortest exchange times. Indeed, in figure 8 the authors shows a comparison between results obtained with the solvent-quenched H/D techniques and the more conventional “TROSY” technique, but this is for protons exhibiting very slow exchange (hours…). Of course, in this figure, the choice of the curves is driven by the comparison with the TROSY method, not suitable for protons in relatively fast exchange. But it is of interest to have an idea of the limit of the solvent-quenched technique in the fast exchange limit: minutes? seconds? Should it be combined to stop-flow techniques? Also, kinetic curves for protons in rapid exchange should be shown in an additional figure, in order to have an idea of the quality of the experimental data.

Author Response

Reviewer 2

  1. Whether pulse-labeling NMR techniques are efficient to highlight protein folding intermediates, there are others techniques that should be mentioned. Even if the authors want to restrict their review to the NMR methods (which is reasonable!), the considerable contribution of High-Pressure NMR to the field in the last decade should be cited. The authors can choose some seminal works like: Kitahara and Akasaka PNAS 100, 3167-72, 2003, or Roche et al., PNAS 109, 6945-50, 2012, or Fossat et al., BJ 111, 2368-76, 2016. Instead, they can also cite recent reviews like Roche et al., Prog. NMR Spectr. 102-103, 15-31, 2017 or Dubois et al., Molecules 25, 5551, 2020. Also, they must mention the work of Fuentes and Wand Biochemistry 37, 9877-9883, 1998, who combined H/D exchange and HP-NMR to investigate protein folding.

We thank the reviewer for the valuable comment. Following reviewer's comment, we chose (Kitahara and Akasaka PNAS 100, 3167-72, 2003) and (Fuentes and Wand Biochemistry 37, 9877-9883, 1998), refs. 195 and 18, respectively, because these two articles are closely related to the present work, and report the pressure-induced ubiquitin unfolding and the native-state H/D exchange monitored by 2D NMR under pressure perturbation. As to the other four articles mentioned by the reviewer, they are all very important contributions to the protein folding studies, but they do not directly related to the present work on the DMSO-quenched 2D NMR spectroscopy, so that these four articles are not cited.

  1. This technique is supposed to be very efficient for fast-exchanging protons, but I cannot find any information on the time-scale covered, especially for the shortest exchange times. Indeed, in figure 8 the authors shows a comparison between results obtained with the solvent-quenched H/D techniques and the more conventional “TROSY” technique, but this is for protons exhibiting very slow exchange (hours…). Of course, in this figure, the choice of the curves is driven by the comparison with the TROSY method, not suitable for protons in relatively fast exchange. But it is of interest to have an idea of the limit of the solvent-quenched technique in the fast exchange limit: minutes? seconds? Should it be combined to stop-flow techniques? Also, kinetic curves for protons in rapid exchange should be shown in an additional figure, in order to have an idea of the quality of the experimental data.

We thank the reviewer for this important comment. Following the comment, we described the expected time scale of the present H/D-exchange technique (lines 217–221). The kinetic curves for four NH protons with different exchange times are provided in Supplementary Figure 1.

Reviewer 3 Report

In this manuscript, Kuwajima et al. attempt to review advance in the field of H/D-Exchange 2D NMR spectroscopy, focusing on a DMSO-quenched method that was developed in past studies. Despite the efforts of the authors, I find that the selected format of the authors does not suit an informative review article and is not particularly suited for the focus of this special issue. The main aspect of the review article is narrow and not of broad interest and, if I could add, a bit outdated to validate the need of a review article. The latter is very evident from the selected references, which are almost without exception at least 4-5 years old, as is even one of the two case studies described. There have been extensive studies on amyloid fibrils or the study of folding intermediates with this approach, which I feel was brushed over in only a short paragraph, whereas the authors spend 8(!) pages to describe 2 personal older cases instead. As such, in this current format, I do not think this manuscript will be of great interest to the reading audience and can not recommend it for publication.

Author Response

Reviewer 3

  1. In this manuscript, Kuwajima et al. attempt to review advance in the field of H/D-Exchange 2D NMR spectroscopy, focusing on a DMSO-quenched method that was developed in past studies. Despite the efforts of the authors, I find that the selected format of the authors does not suit an informative review article and is not particularly suited for the focus of this special issue. The main aspect of the review article is narrow and not of broad interest and, if I could add, a bit outdated to validate the need of a review article. The latter is very evident from the selected references, which are almost without exception at least 4-5 years old, as is even one of the two case studies described.

This special issue is dedicated to the memory of the late Professor Chris Dobson, who made outstanding contributions to the advancement of studies of protein folding and the related areas. One of his major contributions to this area is a series of powerful and extensive studies on equilibrium and kinetic folding intermediates by the H/D-exchange labeling 2D NMR techniques, and these studies were mostly published in 1990s–early 2000s (please see refs. 28–30, 33, 44, 45, 51, 59–63, 127, and 174). A reason why the authors chose the subject entitled "DMSO-quenched H/D-Exchange 2D NMR Spectroscopy and Its Applications in Protein Science" is because we recognize very highly the above contributions of Chris in the field of protein folding. The amyloid studies also constitute his major contribution, but we already have 8 articles in this Special Issue. The novelty is an important factor of review articles, but the brief historical review is also very useful in understanding current ongoing sciences. Considering reviewer's comment, we updated certain references related to protein folding (refs. 21, 176, 177, 184).

  1. There have been extensive studies on amyloid fibrils or the study of folding intermediates with this approach, which I feel was brushed over in only a short paragraph, whereas the authors spend 8(!) pages to describe 2 personal older cases instead. As such, in this current format, I do not think this manuscript will be of great interest to the reading audience and can not recommend it for publication.

Because the study on amyloid fibrils is not a major subject of this article, we only briefly described the amyloid studies in which the DMSO-quenched H/D-exchange method was employed. Nevertheless, we cited a number of references of DMSO-related amyloid studies (ref. 99–134), which may be useful for people who are interested in amyloid studies. Our improved DMSO-quenched H/D-exchanged method with the use of spin desalting columns has currently been used only by us, although the method should be very useful in studies of protein H/D exchange; it made it possible to study the H/D exchange of proteins in concentrated salt or denaturant by 2D NMR. We believe that this article is useful for the audience in the field of protein science.

Round 2

Reviewer 3 Report

I am aware of the early work of Chris Dobson on H/D exchange 2D NMR, but the relevance of the manuscript to the SI was never in question. Nor was the importance of the DMSO-quenched developed protocols. As a review, it can contain older work, but in this case, the manuscript is long and mainly a representation of much older work.

In general, informative reviews should make an effort to shape recent advancement in a field, which is not the case here. Less than 10% of the references are of the last 5 years (in a total of 200! refs), when this number should be at least above 50%. As mentioned by the journal itself, reviews should provide concise and precise updates on the latest progress in the field. Even one of the two cases described, which takes 9 pages in contrast of 5 for the rest of the manuscript - thus narrowing the focus and making it unnecessarily long, is almost 10 years old.

Author Response

In general, informative reviews should make an effort to shape recent advancement in a field, which is not the case here. Less than 10% of the references are of the last 5 years (in a total of 200! refs), when this number should be at least above 50%. As mentioned by the journal itself, reviews should provide concise and precise updates on the latest progress in the field. Even one of the two cases described, which takes 9 pages in contrast of 5 for the rest of the manuscript - thus narrowing the focus and making it unnecessarily long, is almost 10 years old.

We thank the reviewer for the valuable comment to improve our review article. Following the reviewer’s comment, we have removed the portion of “Co-chaperonin GroES”, which may be too old, as pointed out by the reviewer. The manuscript has been extensively shortened, from 16 pages to 11 pages, and we believe that it became much more readable. The number of references published within the last 5 years is 13, and it is still less than 10% of the total number (175) of references. However, this was mainly caused by many citations of historically important papers in Introduction section, concerning the quenched H/D-exchange studies, which gave the basis of the DMSO-quenched method. We believe that the re-revised version is now suitable for publication in the SI of Molecules.